# IHF and Fis as *Escherichia coli* Cell Cycle Regulators: Activation of the Replication Origin *oriC* and the Regulatory Cycle of the DnaA Initiator

**DOI:** 10.3390/ijms241411572

**Published:** 2023-07-18

**Authors:** Kazutoshi Kasho, Shogo Ozaki, Tsutomu Katayama

**Affiliations:** Department of Molecular Biology, Graduate School of Pharmaceutical Sciences, Kyushu University, Fukuoka 812-8582, Japan; shogo.ozaki@phar.kyushu-u.ac.jp (S.O.); katayama@phar.kyushu-u.ac.jp (T.K.)

**Keywords:** DNA replication, initiation, DnaA, *oriC*, nucleoid associated proteins, IHF, Fis, *Escherichia coli*

## Abstract

This review summarizes current knowledge about the mechanisms of timely binding and dissociation of two nucleoid proteins, IHF and Fis, which play fundamental roles in the initiation of chromosomal DNA replication in *Escherichia coli*. Replication is initiated from a unique replication origin called *oriC* and is tightly regulated so that it occurs only once per cell cycle. The timing of replication initiation at *oriC* is rigidly controlled by the timely binding of the initiator protein DnaA and IHF to *oriC*. The first part of this review presents up-to-date knowledge about the timely stabilization of *oriC*-IHF binding at *oriC* during replication initiation. Recent advances in our understanding of the genome-wide profile of cell cycle-coordinated IHF binding have revealed the *oriC*-specific stabilization of IHF binding by ATP-DnaA oligomers at *oriC* and by an initiation-specific IHF binding consensus sequence at *oriC*. The second part of this review summarizes the mechanism of the timely regulation of DnaA activity via the chromosomal loci *DARS2* (DnaA-reactivating sequence 2) and *datA*. The timing of replication initiation at *oriC* is controlled predominantly by the phosphorylated form of the adenosine nucleotide bound to DnaA, i.e., ATP-DnaA, but not ADP-ADP, is competent for initiation. Before initiation, *DARS2* increases the level of ATP-DnaA by stimulating the exchange of ADP for ATP on DnaA. This *DARS2* function is activated by the site-specific and timely binding of both IHF and Fis within *DARS2*. After initiation, another chromosomal locus, *datA*, which inactivates ATP-DnaA by stimulating ATP hydrolysis, is activated by the timely binding of IHF. A recent study has shown that ATP-DnaA oligomers formed at *DARS2*-Fis binding sites competitively dissociate Fis via negative feedback, whereas IHF regulation at *DARS2* and *datA* still remains to be investigated. This review summarizes the current knowledge about the specific role of IHF and Fis in the regulation of replication initiation and proposes a mechanism for the regulation of timely IHF binding and dissociation at *DARS2* and *datA*.

## 1. Introduction

The *E. coli* chromosome is a circular double-strand 4.6 Mb DNA polymer. It forms a condensed nucleoid structure by binding to numerous nucleoid-associated proteins (NAPs) and forming a highly ordered DNA superstructure, such as supercoiled DNA [1,2,3]. The *E. coli* chromosome has a single replication origin, *oriC*, and is precisely replicated only once per cell cycle (Figure 1A,B) [4,5,6,7]. The timing of initiation is coordinated with cellular growth conditions, and even in rapidly growing cells, initiation occurs simultaneously at two or more sister *oriCs* only once at a specific time during the cell cycle.

This review focuses on two NAPs, IHF (integration host factor) and Fis (factor for inversion stimulation), both of which are known to regulate chromosome conformation by modulating DNA supercoiling, transcription of hundreds of genes, and initiation of chromosomal DNA replication [3,4]. IHF exists as a heterodimer of α- and β-subunits and is abundant in *E. coli* cells regardless of the growth phase (5000–10,000 molecules per cell) [3,8]. IHF specifically binds to the consensus sequence (TAAnnnnTTGATW, where W is A or T) and induces sharp DNA bending [9,10]. By contrast, Fis exists as a homodimer and is abundant only in log phase cells (50,000–100,000 molecules per cell), whereas it is scarce in stationary phase cells (<100 molecules per cell) [3,8,11]. The amount of Fis also varies according to changes in growth conditions such as nutrition upshift or the stringent response [11,12]. The balance between activation and autoregulation of *Fis* gene transcription and rapid degradation of Fis protein in the stationary phase results in the temporal abundance of Fis in the log phase [11,13]. Fis specifically binds to its consensus sequence (GnnYAnnnnTRnnC, where Y is T or C and R is A or G), causing shallow curvature, and is a global transcription regulator that interacts with RNA polymerase [14,15]. To achieve precise regulation of replication initiation timing, IHF plays an essential role in the timely activation of *oriC* [16,17,18,19,20] and timely regulation of the activity of the initiator protein DnaA via the chromosomal loci *DARS2* and *datA* (Figure 1A–C) [21,22]. Fis is also essential for activating DnaA via *DARS2* [21]. IHF binding and dissociation at *oriC*, *datA*, and *DARS2* are regulated to occur at precise times during the cell cycle, and Fis binding/dissociation at *DARS2* occurs precisely at the pre-initiation stage [21,22]. Thus, cell cycle-coordinated regulation of timely binding/dissociation of IHF and Fis is crucial for the regulation of replication initiation.

## 2. Initiation Complex at *oriC*

The initiation of DNA replication is a highly regulated process that ensures the accurate and timely duplication of the genome [4,6,7]. In *E. coli*, this process begins with DnaA binding to *oriC,* which is located at 84.6 min on the circular chromosome. The *oriC* region comprises a duplex-unwinding element (DUE) and a flanking DnaA oligomerization region (DOR) containing a cluster of DnaA-binding sites (DnaA boxes) (Figure 1C and Figure 2A). ATP-bound DnaA proteins in concert with IHF form a highly ordered nucleoprotein complex with DOR, resulting in initiation complex formation (Figure 2B) [7,17,23]. By making use of the thermal energy and topological constraint of supercoiled DNA [24,25], the initiation complex transiently unwinds the DUE, and the unwound DUE is stabilized by direct interaction with DOR-bound DnaA, which allows for subsequent loading of the replicative DNA helicase DnaB onto the resulting single-stranded DUE and AT-cluster regions that flank the DUE just outside of the minimal *oriC* region [23,26,27]. The DNA-loaded DnaB helicase further expands the single-stranded DNA regions, thereby initiating the semi-conservative synthesis of new DNA strands by the DNA polymerase III holoenzyme [28].

*E. coli* DnaA is composed of four distinct domains [4,6,29,30]. The N-terminal domain I is involved in protein–protein interactions [31,32,33]. The Phe46 residue of this domain acts as a primary binding site for DnaB and DiaA, a homo-tetrameric protein that interacts with a few DnaA molecules simultaneously, stimulating the assembly of ATP-DnaA molecules on *oriC* [32,34,35]. Domain II is a flexible linker connecting domains I and III [31,36]. Domain III contains AAA+ (ATPase associated with various cellular activities) motifs involved in ATP/ADP binding, ATP hydrolysis, and domain III–domain III interactions [37]. The arginine finger motif Arg285 within this domain interacts with ATP bound to the flanking DnaA protomer, promoting cooperative ATP-DnaA binding to the DOR in a head-to-tail orientation [17,38,39,40]. Moreover, H/B-motifs (hydrophobic Val211 and basic Arg245) in this domain bind single-stranded DUEs (ssDUEs) during DUE unwinding processes [23,24,41]. Recent studies have highlighted the specific role of the His136 residue of domain III in DnaB loading onto ssDUE, consistent with the structural insight that the DnaA His136 residue is exposed to the surface and potentially interacts with other proteins [26,27,42]. Domain IV binds specifically to the DnaA box with an asymmetric 9-mer consensus sequence, TTAWnCACA [43]. These domain architectures are highly conserved among bacterial DnaA family proteins.

The general feature of bacterial DORs is the clustering of multiple DnaA boxes that encode instructions for the formation of the initiation complex. In *E. coli,* the DOR is subdivided into three subregions: left, middle, and right DORs (Figure 2A) [23,44,45,46,47]. The left DOR, comprising six DnaA boxes (R1, τ1, R5M, τ2, I1, and I2) and a specific IHF binding site (IBS1), is strictly required for both DUE unwinding and DnaB loading. In addition, the middle–right DORs with a single DnaA box R2 and the right DOR with five DnaA boxes (R4, C1-3, and I3) increase DnaB loading activity to the maximal level [23,48]. DnaA box τ1 partially overlaps with IBS1 and stimulates DUE unwinding in the presence of HU, but not IHF (see below) [49]. In the presence of IHF, the left DOR forms a dynamic complex comprising IBS1-bound IHF and an ATP-DnaA pentamer bound to R1, R5M, τ2, I1, and I2. This complex promotes DUE unwinding, and the upper strand of the resulting ssDUE bearing TT[A/G]T(T) motifs is subsequently recruited to ATP-DnaA pentamers through IHF binding-induced DOR bending (Figure 2B) [17,24,48]. This ssDUE recruitment mechanism stabilizes the unwound state of DUE, allowing for DnaB loading onto the ssDUE region. Importantly, the DnaA boxes of the left DOR share the same orientation, while those of the middle and right DORs share the opposite orientation. The directional arrangement of the DnaA boxes in the left and right DORs facilitates the head-to-tail oligomerization of ATP-DnaA proteins, which leads to the formation of a pair of ATP-DnaA pentamers bound to the left and right DORs, with the two facing each other. DnaA box R2 in the middle-DOR assists DnaA assembly at the left/right DORs, potentially via a domain I-mediated DnaA-DnaA interaction [45,48]. Spaces from the DnaA box R2 to I2 and C3 are long enough to prevent domain III-domain III interaction. In addition, previous studies have suggested that Fis can bind to a specific site between R2 and C3 in *oriC*, but this specific *oriC*–Fis binding does not, or may only slightly, influence the initiation of replication [20,50,51,52,53]. A recent review summarizes the roles and mechanisms suggested for the Fis-*oriC* interaction [7].

The DUE contains three AT-rich 13-mer repeats, designated as L, M, and R (Figure 1C and Figure 2A). Among these, the M- and R-DUEs are essential for basal DUE unwinding, and the L-DUE enhances the extent of the DUE unwinding and subsequent DnaB loading processes [23]. The upper strand of the M- and R-DUEs contains TTGT and TTATT motifs, which are described collectively as TT[A/G]T(T) motifs (Figure 1C) [4,24]. Upon initial transient unwinding induced by thermal energy and superhelical tension, the single strand of these motifs directly binds to the ATP-DnaA pentamer formed on the left DOR via sharp DNA bending induced by bound IHF, stabilizing the unwound state of the M- and R-DUEs (Figure 2C). This is termed the ssDUE recruitment mechanism.

Recently, HU protein, which is a ubiquitous IHF homolog without DNA-binding sequence specificity, was found to bind the *oriC* site specifically, depending on ATP-DnaA complex formation on the left DOR, which promotes DUE unwinding in a manner principally similar to the IHF-dependent ssDUE recruitment mechanism [49]. The DnaA box τ1 plays a stimulatory role in DUE unwinding, specifically in the presence of HU, but not IHF. HU has an extraordinarily high affinity for bent DNA [54]. ATP-DnaA binding to the DnaA box τ1 stimulates ATP-DnaA complex formation on the left DOR, which would enhance DNA bending in the IBS1 region, thereby stimulating binding of HU to the bent region and DUE unwinding by the ssDUE recruitment mechanism (Figure 2B). Whereas IHF is conserved in only a limited number of phyla, such as proteobacteria, HU protein is ubiquitously conserved in the bacterial domain. Thus, these features support the idea that the ssDUE recruitment mechanism is conserved among diverse bacterial species. Consistently, DnaA-ssDUE interactions have been implicated in other bacterial species including *Thermotoga maritima*, *Bacillus subtilis*, and *Helicobacter pylori* [55,56,57,58]. In *B. subtilis*, the HU homologue HBsu is essential for in vivo replication initiation from *oriC*, and *E. coli* HU stimulates the in vitro DUE unwinding of *B. subtilis oriC* [59,60]. These are consistent with the idea that DNA bending is essential for initiation from *oriC*, and thus the ssDUE recruitment mechanism could be conserved in *B. subtilis*, although specific binding of HBsu to *B. subtilis oriC* remains to be elucidated. In *T. maritima oriC*, ATP-DnaA-dependent site-specific binding of HU is indicated in vitro [49]. Although *oriC* structures suggest the possibility of the involvement of cognate HUs in other species, this possibility remains to be examined experimentally.

The upper stand of L-DUE also contains the TTATT sequence. While the unwinding of the L-DUE is less stable than that of M- and R-DUEs, the right DOR-bound ATP-DnaA complexes engage single-stranded L-DUE, which further stabilizes the unwound state of the DUE, thereby facilitating DnaB loading (Figure 2B) [23]. Upon DnaB loading, the single-stranded region further extends to an AT-rich 12-mer (AT-cluster) that flanks the L-DUE (Figure 2A) [23,28]. In addition, the AT-cluster region enhances the robustness of DnaB loading processes, as demonstrated by studies showing that ATP-DnaA-IHF subcomplexes formed on a mutant *oriC* lacking middle and right DORs require the AT-cluster region for substantial DnaB loading activity in an in vitro reconstituted system [23,48]. This suggests that the AT-cluster supports DnaB helicase loading, likely because it is susceptible to denaturation. Consistently, even in vivo, the AT-cluster−L-DUE regions support replication initiation in cells lacking the right DOR within the *oriC* [61].

## 3. Regulations of *oriC* and DnaA

### 3.1. Replication-Coupled Negative Regulation of oriC: The Sequestration System

In *E. coli*, replication initiation is regulated by various regulatory systems during the cell cycle. The main targets of these systems are *oriC* and DnaA. At *oriC*, the sequestration system comprising Dam methyltransferase and the SeqA hemi-methylated DNA-binding protein mainly regulates the initiation activity of *oriC* in a manner coupled with DNA replication (Figure 2A) [25,62,63,64]. Dam methyltransferase interacts with the GATC sequence to methylate the A residue. As GATC is a palindromic sequence, both A residues in the DNA duplex of this site are methylated, and semi-conservative replication produces two sister DNAs with hemi-methylated GATC sites, which are the binding target of SeqA [65,66,67]. The *oriC* region contains eleven GATC sites and of these, ten sites are concentrated within the DUE and Left-DOR (Figure 2A). SeqA binds to these regions and forms multimers, which alter superhelicity and inhibit DnaA binding, preventing replication initiation [64,68,69,70]. Because SeqA binds to hemi-methylated *oriC* only for a short period after replication of *oriC* during the cell cycle [71,72], regulatory systems for DnaA are also required to prevent untimely extra initiations.

### 3.2. DnaA Cycle: RIDA, DDAH, DARS1/2

#### 3.2.1. Replication-Coupled Negative Regulation of DnaA: The RIDA System

The main system for the inactivation of DnaA is called RIDA (regulatory inactivation of DnaA), which is coupled with DNA replication (Figure 1B) [4,25,29]. The cellular level of ATP-DnaA, the initiation-active form of DnaA, fluctuates during the cell cycle with a single peak at the time of chromosomal replication initiation. RIDA is required to reduce the ATP-DnaA levels. If RIDA is inactivated, a high ATP-DnaA level is maintained, resulting in severe extra initiations and extra copies of the chromosome, which inhibits cell division and cell growth.

In RIDA, the DNA-loaded form of the clamp (β subunit) of the DNA polymerase III holoenzyme is the key element [73]. After initiation, replisomes, including DNA polymerase III holoenzyme, are assembled and replicate the leading and lagging strands. The clamp is loaded onto both strands of DNA and binds to the core complex of the holoenzyme, supporting the processive synthesis of long DNA regions. During DNA replication of the lagging strand, the clamp remains loaded onto the nascent DNA region after Okazaki fragment synthesis [74]. This DNA-loaded form of the clamp then interacts with the ADP form of Hda, an AAA+ protein with a short N-terminus bearing a clamp-binding site [73,75,76,77,78]. The DNA-clamp-Hda complex interacts catalytically with ATP-DnaA, promoting ATP hydrolysis for the generation of ADP-DnaA. The arginine finger motif of Hda is required for the hydrolysis of DnaA-bound ATP [79]. In the next round of Okazaki fragment synthesis, another clamp molecule was loaded onto the replisome. Thus, RIDA activation is DNA replication-dependent, which ensures the timely inactivation of DnaA. The principle of RIDA, the DNA-loaded clamp-dependent inactivation of the initiating protein, is conserved in prokaryotic and eukaryotic cells [29,80,81,82]. Hda is activated by the binding of ADP [76,79], but how this binding is regulated in vivo is still a mystery.

#### 3.2.2. IHF-Dependent Negative Regulation of DnaA: The DDAH System

RIDA is backed up by another system termed DDAH (*datA*-dependent DnaA-ATP hydrolysis), which also efficiently inactivates ATP-DnaA (Figure 1B) [22,83]. *datA* is located at 94.7 min on the *E. coli* chromosome, near *oriC* (Figure 1A) [84,85]. Deletion of *datA* causes untimely initiations in growing cells [84,86]. The minimal *datA* region contains a single IBS and four DnaA boxes, of which three (DnaA boxes 2, 3, and 7) are essential and one (DnaA box 4) is stimulatory for *datA* function (Figure 1C) [83]. Spacing between IBS and the essential DnaA boxes is important for *datA* function [22,86]. The IHF-*datA* complex interacts catalytically with ATP-DnaA, promoting the hydrolysis of DnaA-bound ATP [22,83]. As IHF sharply bends DNA, DNA loop formation is postulated to promote interactions between DnaA molecules bound to DnaA boxes 2 and 3, which is the key event in the activation of ATP hydrolysis. Notably, DnaA-ATP hydrolysis was not observed at *oriC* [22]. Thus, IHF-dependent DnaA-DnaA interactions at the *datA* locus specifically support DDAH.

The timely regulation of DDAH depends on the timely binding of IHF to *datA* [22]. IHF binds to *datA* only for a short period after initiation, ensuring the timely activation of DDAH. How this binding is regulated in vivo in a timely manner is still unknown. In addition, chromosomal positional effects of *datA* have been reported. When the locus of *datA* is moved to a position distant from *oriC*, the *datA* function is moderately inhibited, resulting in precocious initiations [84,87]. The mechanisms responsible for this effect remain to be investigated. The relative dosage of *datA* on the replicating chromosome could be important for efficient *datA* function. Alternatively, the chromosomal position of *datA* could affect the interaction of *datA* with IHF or DnaA, depending on the subcellular position of the *datA* site. The *datA* site is present in the tightly folded ORI macrodomain of the chromosome, and analysis of its subcellular localization suggests that it is located near *oriC* [88,89,90].

#### 3.2.3. IHF- and Fis-Dependent Positive Regulation of DnaA: The DARS System

ADP-DnaA molecules are converted to ATP-DnaA by a regulatory system called DARS. This system depends on *DARS1* and *DARS2*, which are specific chromosomal regions that possess a specific set of DnaA boxes, i.e., DnaA boxes I, II, and III, forming the common core region (Figure 1C) [21,91,92]. In the DARS system, ADP dissociates from ADP-DnaA present in the *DARS1/DARS2* complexes, resulting in the release of apo-DnaA and its binding to ATP, the active form of DnaA. *DARS2* plays a predominant role in the DARS system in vivo [21,91], carries specific binding sites for IHF and Fis, and is timely activated during the cell cycle in a manner depending on the temporal binding of IHF and Fis to IBS1-2 and FBS2-3 (Figure 1C) [21].

Specific arrangements of the three DnaA boxes in *DARSs* are essential for ADP dissociation from ADP-DnaA. Unlike the head-to-tail arrangements of DnaA boxes at *oriC*, the head-to-head arrangement of DnaA boxes I and II is a characteristic of *DARSs* and is functionally essential [91,92]: ADP-DnaA molecules bound to the two sites are suggested to interact with each other through their AAA+ domains, causing specific structural changes of ADP-DnaA to dissociate ADP. DnaA-bound DnaA box III stimulates these reactions by interacting with DnaA bound to DnaA box II and enhancing overall complex formation. The assembly of DnaA molecules on *DARSs* can also be stimulated in vitro by DiaA [92,93], a specific stimulator of ATP-DnaA assembly at *oriC*. ADP dissociation generates apo-DnaA molecules, which are labile in and released from these complexes. At *DARS2*, IHF and Fis could promote specific interactions between DnaA molecules bound to the essential DnaA boxes. Sequences similar to those of *DARS1* and *DARS2* are conserved in the genomes of many proteobacterial species [91]. In addition, similar to *datA*, chromosomal positional effects are known for *DARS1* and *DARS2* [87,94]. As an additional note, acidic phospholipids are suggested to mediate ATP-DnaA production from ADP-DnaA; however, its biological significance as well as the cell cycle-coordinated regulation remains to be further explored [95].

## 4. Cell Cycle-Coordinated Regulation of IHF/Fis Binding and Dissociation

### 4.1. Regulation of oriC-IHF Binding

During the *E. coli* cell cycle, IHF stably binds to *oriC* specifically at the stage of replication initiation and dissociates from *oriC* immediately after initiation [16,18,22]. Further characterization of the genomic IHF binding pattern using GeF-seq (genome footprinting with high-throughput sequencing) at base-pair resolution has provided evidence for the unique and specific regulation of *oriC*-IHF binding stability at the initiation stage [18,96]. IHF recognizes a 33-mer pre-initiation stage-specific IHF binding consensus sequence 1-GTTGnnGnnnWnnAAAnnCAnnnnTTTnWnAAC-33 that consists of the core DNA elements “19-CA-20” and “25-TT-26” which are conserved with conventional IBS consensus WATCAAnnnnTTR, as well as unique surrounding elements “1-GTTG-4” and “31-AAC-33” (Figure 2A). Consistently, IHF binding affinity to each IBS depends on the surrounding A/T elements [9], and molecular dynamics simulations of IHF binding to supercoiled plasmid DNA have suggested that DNA supercoiling affects DNA recognition by IHF and converts the IHF-DNA complex into a more wrapped state [10]. This suggests that at a specific cell cycle stage, a global change in the higher-order genomic structure may increase the dependency on unique surrounding DNA elements in the IHF binding consensus sequence and stabilize IHF binding at specific genomic loci including *oriC*.

The secondary IBS at *oriC* (IBS2) with the initiation-specific consensus sequence overlaps with the DnaA box R1 (Figure 2A) [18], suggesting that IHF binds to either the primary IBS (named IBS1) or IBS2, and that if DnaA occupies R1, IHF can no longer interact with IBS2 and will preferentially bind to IBS1. Another possibility is that IBS2 may act as a reservoir of IHF to prevent the free diffusion of IHF from IBS1 and maximize *oriC*-IHF binding at initiation. This is consistent with IHF binding modes; IHF-DNA binding and bending occur in a stepwise manner, in which the reaction rate of IHF-DNA binding without DNA bending is rapid, and that of IHF-induced DNA bending is much slower and rate-limiting [97]. Initiation stage-specific stabilization of *oriC*-IHF binding requires *oriC* DnaA box R1, whereas the chromosomal *oriC* position is irrelevant to stabilization at the initiation stage [18]. In the initiation complexes, ATP-DnaA forms specific bulky oligomers on *oriC* [17,23,45], thus stabilizing the bending of DNA bound to IHF in the IBS-proximal region and preventing IHF dissociation from *oriC*. These unique features of IHF binding dynamics at *oriC* shed light on the sophisticated mechanism that ensures timely IHF binding for strict regulation of replication initiation.

### 4.2. Regulation of DARS2-Fis Binding and Dissociation

A recent study highlighted the Fis-dependent cell cycle control of *DARS2* activation. During the cell cycle, Fis temporarily binds to *DARS2* FBS2-3 at the pre-initiation stage to produce ATP-DnaA and dissociates at the initiation stage (Figure 3A) [93]. For timely regulation of Fis binding, the high affinity DnaA box V and surrounding low-affinity boxes V-a, V-b, and V-c that overlap with FBS2-3 are required (Figure 3B). Three or four molecules of ATP-DnaA, but not ADP-DnaA, form head-to-tail oligomers on clusters of DnaA boxes centered at DnaA box V and directly dissociate Fis from FBS2-3, resulting in cell cycle-dependent *DARS2* inactivation via negative feedback. This simple negative feedback regulation of *DARS2* is fundamental for the timely repression of *DARS2* activity, which is required for the optimal regulation of cellular ATP-DnaA levels and replication initiation in a manner redundant to IHF regulation. ATP-DnaA assembly on the DnaA box cluster as well as Fis dissociation are stimulated by DiaA; however, it is currently unclear how DiaA binding to ATP-DnaA at the *DARS2* locus is regulated during the cell cycle.

In the growth phase regulation, the cellular level of Fis peaks in the exponential phase [3,8] and Fis binds to *DARS2* FBS2-3, whereas it no longer binds in the stationary phase where Fis is almost absent in cells [21]. This means that in addition to the cell cycle timing, *DARS2* monitors cellular growth phases via Fis for efficient replication initiation in rapidly growing cells.

### 4.3. Regulation of DARS2-IHF Binding

The timing of IHF binding to and dissociation from *DARS2* is strictly regulated in a cell cycle-coordinated manner [21]. Contrary to the *datA* locus, IHF binds to *DARS2* IBS1-2 specifically at the pre-initiation stage of the cell cycle to produce ATP-DnaA for initiation at *oriC* [21]. Consistently, overexpression of IHF in cells with a *DARS2* mutation results in constitutive Fis binding at *DARS2* FBS2-3, which causes excessive initiations [93]. This stresses the importance of cell cycle-coordinated IHF dissociation from *DARS2* for the regulation of the initiation of DNA replication (Figure 3A). Regulation of *DARS2*–IHF binding/dissociation is resistant to rifampicin [21], suggesting that IHF binding and dissociation at *DARS2* are regulated independently of transcription. The timing of *DARS2*–IHF binding is also independent of replication initiation at *oriC* and replication fork progression [21], supporting the possibility that a specific regulatory factor for timely *DARS2*–IHF binding/dissociation is linked with specific cell cycle events. However, the regulatory mechanism of *DARS2*-IHF binding/dissociation remains totally unknown.

### 4.4. Regulation of datA-IHF Binding

During the cell cycle, IHF temporarily binds to *datA* at the post-initiation stage and dissociates during initiation (Figure 3A) [22]. Since *datA* is located near *oriC* and is duplicated soon after replication initiation [85,87], IHF binding/dissociation at *datA* occurs during ongoing chromosome replication, suggesting that timely *datA*-IHF binding is coupled with certain cell cycle events and is independent of the passage of the replication machinery.

A potential regulator of *datA*-IHF binding is DNA supercoiling. Negative DNA supercoiling, which is controlled by DNA topoisomerase/gyrase and local transcription, stabilizes *datA*-IHF binding in vitro and stimulates DDAH activity [83]. It is known that IHF as well as Fis preferentially bind to curved DNA rather than to straight-shaped DNA [98,99]. Molecular dynamics simulations of IHF binding to supercoiled plasmid DNA have suggested that IHF preferentially binds to its consensus sequence at the top of kinked DNA and pins the supercoiled loop [10]. Consistently, inhibition of negative DNA supercoiling by the DNA gyrase inhibitor novobiocin decreases the ability of *datA* to repress replication initiation in vivo [83]; however, the mechanism responsible for the cell cycle-coordinated regulation of DNA supercoiling at the *datA* locus is unknown.

## 5. Perspectives

### 5.1. Regulatory Mechanism for oriC-IHF Dissociation

In contrast to what is known about the timely regulation of *oriC*-IHF binding, little is known about the mechanism responsible for *oriC*-IHF dissociation after initiation. Several potential mechanisms for the regulation of *oriC*-IHF binding/dissociation have been proposed: (1) passage of the replication machinery, (2) *oriC* methylation and sequestration, and (3) oscillation of the ATP-DnaA level. Concerning the first possibility, after replication initiation, the replication machinery passes through *oriC* to accomplish chromosome duplication; however, it remains unclear whether the replication machinery can directly dissociate ATP-DnaA and IHF from *oriC* or whether other unknown factors are involved. Concerning the second possibility, both *oriC* IBS1 and IBS2 contain the Dam-dependent methylation sequence GATC (Figure 2A); therefore, SeqA-dependent *oriC* sequestration after initiation could potentially prevent IHF binding to *oriC* [4,18,70]. Also, IHF binds with higher affinity to a fully methylated IBS1 fragment than to a hemi- or unmethylated fragment [100], suggesting that the timing of *oriC*-IHF binding could be determined by the status of IBS1 methylation after *oriC* sequestration ends. Concerning the third possibility, the transcriptional level of the *mioC* gene adjacent to *oriC* fluctuates during the cell cycle and is inhibited during pre-initiation by the increase in the ATP-DnaA level [101,102]. After initiation, *mioC* transcription is temporarily activated, and RNA polymerase passes through the whole *oriC* region, suggesting that *mioC* transcription could prevent abortive IHF binding at *oriC*. These or other underlying hypotheses need to be tested to determine whether they can explain the mechanism of IHF dissociation from *oriC*.

### 5.2. Possible Regulatory Mechanisms for IHF/Fis Binding and Dissociation at DARS2 and datA

First, IHF binding/dissociation could be regulated via a synergistic effect with Fis. Systematic analysis of various synthetic transcriptional promoters [103] has revealed that Fis binding at a specific locus can alter the specificity of DNA recognition by IHF to control the transcription activity. Also, simultaneous binding of IHF and Fis can synergistically function to activate the transcription activity, implying that Fis may monitor the *DARS2*-IHF complex to regulate *DARS2*-IHF formation or dissociation at specific times during the cell cycle.

Second, recent in vitro studies have reported that a solution-phase Fis interacts with and dissociates the Fis-DNA complex via two separate mechanisms, namely, facilitated dissociation and cooperative dissociation [104,105]. In facilitated dissociation, DNA-binding domains of the solution-phase Fis symmetrically interact with and dissociate the Fis-DNA complex in a Fis concentration-dependent manner. In cooperative dissociation, the solution-phase Fis directly interacts with the Fis-DNA complex to form binary Fis intermediates via physical contact, leading to dissociation of Fis from the DNA. These Fis exchange reactions have important implications for the regulation of *DARS2* activity as well as for the regulation of ATP-DnaA-dependent dissociation of Fis [93]. Solution-phase Fis or another regulatory protein may directly interact with the Fis-*DARS2* complex to ensure timely Fis dissociation, and *DARS2* FBS4-5, which is located adjacent to FBS2-3 [21], may regulate Fis dissociation via cooperative dissociation, possibly under specific growth conditions, such as poor nutrition or stress.

Third, since the minimal region of *datA* required for full DDAH activity is located in an intergenic region downstream of the tRNA_Gly_ (*glyV-glyX-glyY*) operon between the *yjeV* gene and *queG* gene (Figure 3C) [7,85,86], it is possible that *datA*-IHF binding/dissociation is regulated by transcription of the *datA* locus. Consistently, the transcription inhibitor rifampicin inhibits timely binding and dissociation of *datA*-IHF complexes [22], and *datA* activity is inhibited by the translocation of *datA* to a highly transcribed position [87]. However, the cell cycle-coordinated regulation of *datA*-IHF binding via transcription, DNA supercoiling, or other unknown factors requires further study.

### 5.3. Similar Mechanism of Other DNA Bending Proteins among Species

NAPs are functionally conserved in most histone-deficient organisms or organella, and IHF-like DNA bending proteins are known to regulate DNA replication as well as transcription and chromosome packaging in various organisms, which include the following: HU, an IHF homolog in most bacteria [3,19,106]; mitochondrial Abf1 and Abf2 (autonomously replicating sequence-binding factor 1 and 2) in yeast [107,108,109]; mitochondrial HMG (high mobility group) family protein TFAM (mitochondrial transcription factor A), a eukaryotic equivalent of NAPs [108,110]; and MC1 (methanogen chromosomal protein 1) in some classes of methanogenic archaea [3,111]. Genome DNA replication in mitochondria and archaea is also controlled in a cell cycle-coordinated manner [112,113]; however, further work will be needed to reveal the mechanisms regulating the binding of these NAPs to replication origins in mitochondria and archaea. This review assembles new information on the timely regulation of NAP binding/dissociation that should facilitate the understanding of the mechanism of cell cycle-coordinated chromosomal replication. The dissociation of *E. coli* HU and the yeast HMG family protein NHP6A from DNA can be mediated by protein exchange reactions as seen for Fis, as described in Chapter 4–2 [114], implying that the dissociation of NAPs and their equivalents is regulated by a common mechanism. Also, DNA super-structures, such as DNA supercoils and G4 DNA, are commonly related to DNA binding, bending, and wrapping of NAPs, e.g., *E. coli* HU, *Thermotoga maritima* HU, *Streptomyces coelicolor* HU, and *Mycobacterium tuberculosis* NapA preferentially bind to negatively-supercoiled DNA, while *Caulobacter crescentus* GapR and eukaryotic HMGA2 preferentially bind to positively-supercoiled DNA [115,116,117,118,119,120]. *E. coli* Hfq prefers G4 DNA [121]. DNA super structure-dependent recognition is not limited to NAPs, but is also a feature of the eukaryotic replication initiator ORC (origin recognition complex), which binds preferably to supercoiled or G4-containing replication origins [122,123]. The information assembled in this review indicates that the cell cycle coordination of the initiation of chromosomal DNA replication and timely binding/dissociation of specific regulatory NAPs, such as IHF and Fis, or their equivalents, determines the timing of initiator protein activation and initiation at origins of replication.

## Figures and Tables

**Figure 1 ijms-24-11572-f001:**
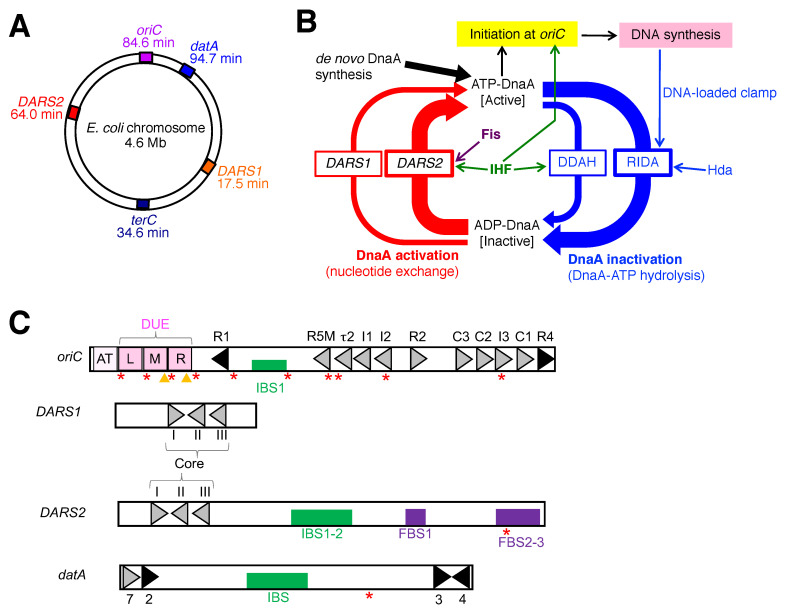
(**A**) Schematic representation of the genomic DNA elements *oriC*, *datA*, *DARS1*, *DARS2*, and *terC* on the 4.6 Mb circular *E. coli* chromosome. Positions of each locus are indicated on the scale of 0–100 min. (**B**) Schematic presentation of the regulatory cycle of DnaA. ATP-DnaA initiates chromosomal DNA replication from *oriC* with the aid of IHF. After initiation, ATP-DnaA is converted to replication-inactive ADP-DnaA (RIDA and DDAH systems). RIDA requires the complex of Hda protein and the DNA-loaded form of the clamp subunit of DNA polymerase III holoenzyme, and DDAH requires the *datA*-IHF complex. Prior to the next round of initiation, the *DARS1* and *DARS2* loci stimulate nucleotide exchange of ADP-DnaA to produce ATP-DnaA. *DARS2*-dependent ATP-DnaA production requires IHF and Fis. (**C**) Structures of *oriC*, *datA*, *DARS1*, and *DARS2*. Open bars indicate the minimal region of each DNA element. Black or gray triangles represent high- or low/middle-affinity DnaA boxes, respectively. Filled squares represent IHF binding sites (IBS; shown in green) and a Fis binding site (FBS; shown in blue). Pink squares represent the duplex-unwinding element (DUE) of *oriC*, including the AT-rich 12 bp element that flanks L-DUE just outside the minimal *oriC*. Both *DARS1* and *DARS2* have core regions containing DnaA boxes I–III. *DARS2* also contains additional DnaA boxes, regulatory IBS, and FBS. Minimal *datA* consists of DnaA boxes 2, 3, and 7 and a single IBS. The *datA* DnaA box 4 stimulates DDAH in vitro. GATC sequences subject to Dam methylation are indicated as red stars. TT[A/G]T(T) motifs in L-, M-, and R-DUEs are shown as orange arrowheads.

**Figure 2 ijms-24-11572-f002:**
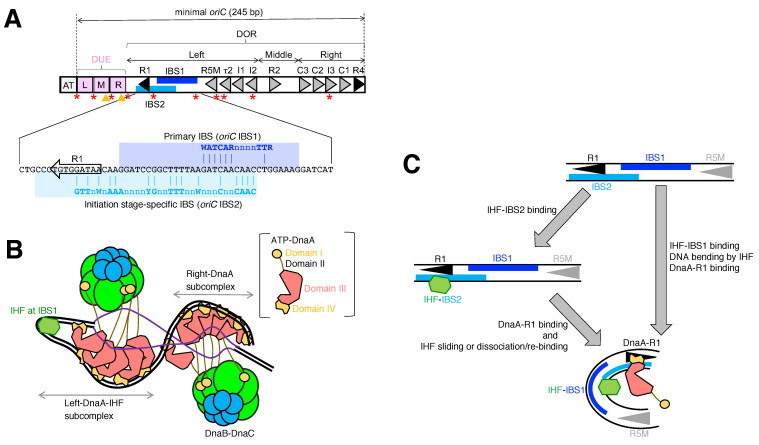
(**A**) Detailed structure of *oriC*. Features of the *oriC* sequence are shown, including DnaA boxes (triangles), primary IBS (IBS1; blue bar), the DUE including the AT-rich 13 bp elements (pink squares), and the left/middle/right DORs. GATC sequences subject to Dam methylation are indicated as red stars. TT[A/G]T(T) motifs in L-, M-, and R-DUEs are shown as orange arrowheads. The sequence of the initiation stage-specific secondary IBS (IBS2; sky blue bar), which overlaps with the DnaA box R1, is shown. (**B**) Model of an unwound state of the initiation complex. Cartoons of DnaA domains I (orange), II (black line), III (red), and IV (brown) are schematically presented. IHF is shown as a light green hexagon. Head-to-tail subcomplexes of ATP-DnaA are formed on the left and right DORs and induce subsequent DnaB loading. (**C**) Model of DnaA box R1-mediated stabilization of *oriC*-IHF binding during the initiation period. First, IHF binds to either IBS1 or IBS2, and when ATP-DnaA occupies DnaA box R1, IHF can no longer bind to IBS2. In the initiation complex, IHF binding and bending at IBS1 are further stabilized by an ATP-DnaA oligomer.

**Figure 3 ijms-24-11572-f003:**
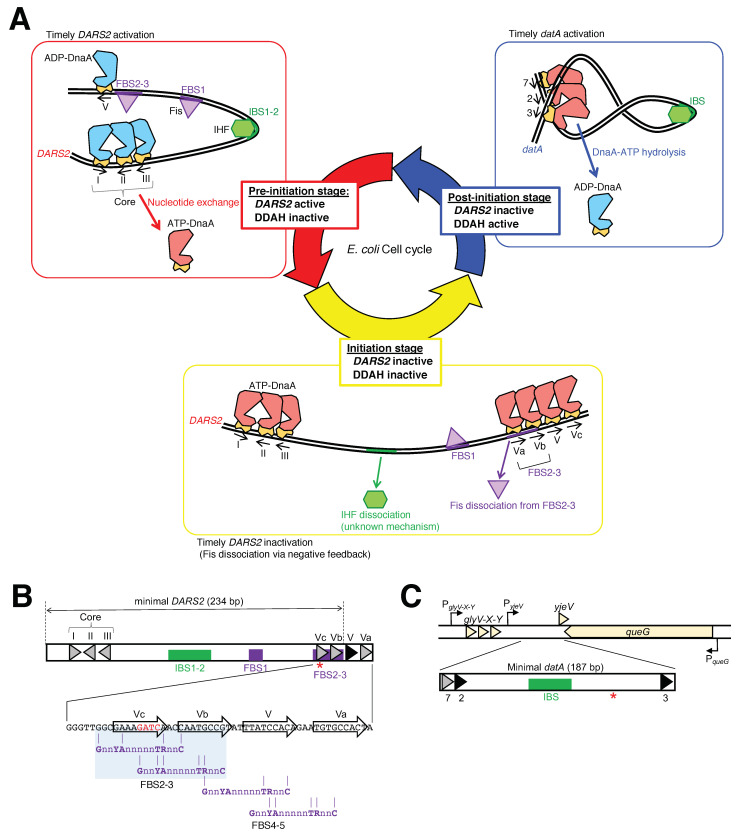
(**A**) A schematic view of the timely binding/dissociation of IHF and Fis at *DARS2* and *datA*. At the pre-initiation stage, during which the ADP-DnaA level is high, the *DARS2*–IHF–Fis complex increases the level of ATP-DnaA. During the pre-initiation stage, ADP-DnaA occupies the DnaA box V, allowing Fis to bind to FBS2–3 to promote *DARS2*-dependent DnaA nucleotide exchange. Next, at the initiation stage, the elevated ATP-DnaA level results in initiation of replication at the IHF-bound *oriC*, at which point IHF and Fis dissociate from *DARS2*. During initiation, ADP-DnaA is replaced with ATP-DnaA oligomers around DnaA box V, thereby competitively restricting Fis access to FBS2-3. Concurrently, IHF dissociates from *DARS2* via an unknown mechanism. After initiation, IHF temporarily binds to *datA* and activates DDAH to inactivate DnaA. For simplicity, domains I-II are omitted in the cartoon of DnaA. (**B**) The sequence of the FBS2–5 and overlapping DnaA boxes. Black arrows indicate the DnaA box consensus sequence (TTWTnCACA). The Fis binding consensus sequence is GnnYAnnnnnTRnnC. GATC sequences subject to Dam methylation are indicated as red stars. (**C**) A schematic view of the genomic *datA* locus. Genes or ORFs (*glyV-X-Y*, *yjeV*, and *queG*) are indicated by orange arrows. Bent arrows indicate transcriptional promoters (P*_glyV-X-Y_*, P*_yjeV_*, and P*_queG_*). The minimal *datA* region (187 bp) is located between *glyV-X-Y* and *queG*.

## Data Availability

Not applicable.

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
