# Peer review of "IHF and Fis as Escherichia coli Cell Cycle Regulators: Activation of the Replication Origin oriC and the Regulatory Cycle of the DnaA Initiator"

_ijms, 2023, doi:10.3390/ijms241411572_

Round 1
Reviewer 1 Report
This review by Kasho et al. covers the roles of the DNA bending proteins, Fis and IHF, in the related processes of initiation of chromosome replication and control of DnaA-ATP levels. The manuscript focuses on how dynamic binding of these nucleoid associated proteins to oriC, DARS, and dat helps maintain precise timing of initiation during the bacterial cell cycle. Overall, the manuscript is well organized and the manuscript is clearly written. However, the manuscript is not yet suitable for publication because it is an incomplete review of the subject: specifically there is a complete absence of any review of the published literature covering binding of Fis to oriC, the effect of this binding on initiation, the dynamics of Fis binding during the cell cycle, or of models describing how the dynamic Fis/oriC interaction affects the timely assembly of the initiation complex in a growth-rate dependent way by modulating binding of IHF and DnaA. This material might fit best in Section 4.1 (Regulation of oriC-IHF binding) but some discussion of Fis in the formation of the initiation complex might reasonably be included in sections 2 and 3. Admittedly, there are conflicting stories in the literature regarding Fis’s role at oriC, but the revised version of this review should include the most relevant/recent studies, address conflicts, and include any previously published models that have proposed resolutions to the conflicts. The binding sites for Fis on oriC also need to be included in the maps in Figures 1 and 2.
There are a few other issues that should be addressed in a revised manuscript:
1) Line 100,104: Reference 44 states that tau1 does not contribute to the initiation complex, and in agreement with this, it is not shown in Figs 1 or 2. The authors should consider removing the reference to this site.
2) Lines 130-137: The wording in this section seems to imply that HU and/or DNA bending plays a role in DnaA binding to ssDUE in the oriCs of B. subtillis and H. pylori, in a way that is similar to the “ssDUE recruitment mechanism” described in E. coli. However, there is no evidence that DNA bending plays a role in these two bacterial types. Rather, the published data indicate that there is a lateral extension of a DnaA-ATP filament that begins with DnaA bound to a high affinity site; DNA bending is not required. The authors should clarify whether or not the ssDUE recruitment model requires a DNA bend, or if it simply means that ssDNA in the DUE is bound by of oligomeric DnaA-ATP. If the latter, it should be noted that DnaA binding to ssDUE as part of an origin unwinding mechanism was reported by Speck and Messer in 2001 (EMBO 20:1469, 2001).
3) Lines 261-275, Fig 2C: According to published literature, R1 is bound by DnaA throughout the cell cycle (reported in ref 16, 19, and 62). Given this, if binding of DnaA to R1 prevents binding of IHF to IBS2, it is hard to see how IBS2 affect binding at IBS1 at any time, or be relevant to initiation timing. The authors should at the least revise this section to include whether persistent R1 binding would reduce the role of IBS2, or remove the section and Fig 2C.
4) Line 345: The appropriate (or at least earlier) reference for mioC transcription in the cell cycle is Theisen et al., Mol Microbiol, 10: 575, 1993.
5) Section 3.2.3, Line 239: there is a brief statement that IHF and Fis could promote specific DnaA interaction within DARS2, but it is unclear whether there are any published studies that might clarify the roles of the binding proteins at DARS2. Have the sites been mutated? A slightly more detailed discussion, even if speculative, could increase the impact of the manuscript.:
Very minor comment: Line 120, the word “strand” is misspelled.
Author Response
Response to Reviewer 1's comments
This review by Kasho et al. covers the roles of the DNA bending proteins, Fis and IHF, in the related processes of initiation of chromosome replication and control of DnaA-ATP levels. The manuscript focuses on how dynamic binding of these nucleoid associated proteins to oriC, DARS, and dat helps maintain precise timing of initiation during the bacterial cell cycle. Overall, the manuscript is well organized and the manuscript is clearly written.
--We appreciate the reviewer for the assessment of our manuscript. All revisions are highlighted in red characters in the revised manuscript. Please see the attachment.
However, the manuscript is not yet suitable for publication because it is an incomplete review of the subject: specifically there is a complete absence of any review of the published literature covering binding of Fis to oriC, the effect of this binding on initiation, the dynamics of Fis binding during the cell cycle, or of models describing how the dynamic Fis/oriC interaction affects the timely assembly of the initiation complex in a growth-rate dependent way by modulating binding of IHF and DnaA. This material might fit best in Section 4.1 (Regulation of oriC-IHF binding) but some discussion of Fis in the formation of the initiation complex might reasonably be included in sections 2 and 3. Admittedly, there are conflicting stories in the literature regarding Fis’s role at oriC, but the revised version of this review should include the most relevant/recent studies, address conflicts, and include any previously published models that have proposed resolutions to the conflicts. The binding sites for Fis on oriC also need to be included in the maps in Figures 1 and 2.
-- Thank you for pointing about possible oriC-Fis interaction. As the reviewer mentioned there are conflicts in interpretations of possible oriC-Fis interaction; 1) a paper by Weigel et al (Mol Microbiol, 2001, 40, 498-507) clearly showed that chromosomal oriC131 mutant (Fis-binding site mutant) has no effect on initiation activity or timing, and 2) a paper by Margulies and Kaguni (J Biol Chem, 1998, 26, 5170-5175) concluded that even though Fis and HU are present in growing cells at a similar level, excessive amounts of Fis are required for inhibition of oriC plasmid replication in vitro (i.e., in the presence of 29 ng HU, at least 150 ng Fis is required for the inhibition), and this is due to sequence-non-specific multiple binding of Fis to DNA, and that sequence-specific primary binding of Fis within oriC has no influence on the initiation process. A paper by Wold et al., (Nucleic Acids Res, 1996, 24, 3527-3532) also provides data showing that the addition of Fis inhibits initiation at oriC in vitro, regardless of the presence or absence of the Fis binding site within oriC.
Thus, these three papers are well consistent with the conclusion that the possible Fis binding site within oriC is not important. Unlike these, 3) papers from Dr. Leonard group suggested possible roles for oriC-Fis binding. i.e., [1] Ryan et al. (Mol Microbiol, 2004, 51, 1347-1359) provided in vitro data that Fis inhibits ATP-DnaA binding to low affinity DnaA boxes, IHF binding and DUE unwinding, and inversely excess ATP-DnaA repress oriC-Fis binding; however, in vivo data supporting these events are not provided. [2] Cassler et al., (EMBO J, 1995, 14, 5833-5841) performed in vivo DMS footprinting with minichromosome and provided the data that oriC-Fis binding is weaker, compared with in vitro oriC-Fis binding, and also oriC-Fis binding is basically constant throughout cell cycle but might be slightly reduced at the time of replication initiation; in this case, Fis is suggested to be an inhibitor of initiation. [3] Kaur et al. (Mol Microbiol, 2014, 91, 1148-1163) provided in vivo data suggesting that when DnaA box R4 is mutated, Fis-binding sequence within oriC stimulates replication initiation by unknown mechanism; in this case, a stimulatory role for Fis is postulated but the significance of Fis binding in oriC-WT cells remains unclear. [4] Rao et al. (Front Microbiol, 2018, 9:1673) provided an indirect suggestion by analysis of oriC DnaA boxes I3/C3 mutant that Fis binding might be inhibited in this mutant, slightly inhibiting initiation; However, as DnaA binding patterns to the whole mutant oriC are not shown in this study, the results can be explained only by indirect inhibition of DnaA binding to the essential site like R5M or by abnormal DnaA complex formation without consideration of Fis binding. Taken all together, we fairly judged that the conflicts are not resolved so far, and further analysis and direct evidence are still required for proving the significance of chromosomal oriC-Fis binding. Also, importantly, the oriC section is NOT the main part of this review. Thus for simplicity and for balance among other sections, we corrected our manuscript minimally and included the sentence as below.
Line 120-125 in the revised manuscript: In addition, previous studies suggested that Fis can bind to a specific site between R2 and C3 in the chromosomal oriC but this specific oriC-Fis binding does not influence the initiation of replication [20,50,51], although there remains a possibility that the binding could slightly influence initiation depending on growth conditions [52,53]. Further experiments are essential to prove the regulatory role of Fis at oriC.
There are a few other issues that should be addressed in a revised manuscript:
1) Line 100,104: Reference 44 states that tau1 does not contribute to the initiation complex, and in agreement with this, it is not shown in Figs 1 or 2. The authors should consider removing the reference to this site.
--Thank you for pointing this out. Yoshida et al., (Nucleic Acids Res, 2023, gkad389) revealed that DnaA box tau1 plays a stimulatory role in DUE unwinding specifically in the presence of HU, but not IHF. Thus contribution of tau1 is clear and the manuscript does NOT need to be corrected. In the revised manuscript, we added new sentences related to this, as below.
Line 106-107 in the revised manuscript; DnaA box t1 partially overlaps with the IBS1 and stimulates DUE unwinding in the presence of HU, but not IHF (see below) [49].
Line 139-143 in the revised manuscript; DnaA box t1 plays a stimulatory role in DUE unwinding specifically in the presence of HU, but not IHF. HU has extra-ordinally high affinity for bent DNA [54]. ATP-DnaA binding to DnaA box t1 stimulates ATP-DnaA complex formation on the left DOR, which would enhance DNA bending at IBS1 region, thereby stimulating binding of HU to the bent region and DUE unwinding by ssDUE recruitment mechanism (Figure 2B).
2) Lines 130-137: The wording in this section seems to imply that HU and/or DNA bending plays a role in DnaA binding to ssDUE in the oriCs of B. subtillis and H. pylori, in a way that is similar to the “ssDUE recruitment mechanism” described in E. coli. However, there is no evidence that DNA bending plays a role in these two bacterial types. Rather, the published data indicate that there is a lateral extension of a DnaA-ATP filament that begins with DnaA bound to a high affinity site; DNA bending is not required. The authors should clarify whether or not the ssDUE recruitment model requires a DNA bend, or if it simply means that ssDNA in the DUE is bound by of oligomeric DnaA-ATP. If the latter, it should be noted that DnaA binding to ssDUE as part of an origin unwinding mechanism was reported by Speck and Messer in 2001 (EMBO 20:1469, 2001).
--Thank you for the comment about initiation mode of other bacteria. Karaboja and Wang., (J Bacteriol, 2022, 204, e0011922) clearly indicated that IHF/HU homologue HBsu is essential for replication initiation from oriC, well consistent with the idea that DNA bending is essential for initiation from oriC and ssDUE recruitment mechanism is present in B. subtilis. The DnaA filament model is suggested by experiments analyzing interaction of DnaA with only artificial forms of DNAs, i.e., pre-existing ssDNA, bubble form DNA or linear DNAs with a gap or a nick. However, unwinding of oriC in supercoiled DNA (which should require HBsu) has not been analyzed in B. subtilis. Thus there remains a possibility that the DnaA filament model is just an artefact seen in specific in vitro conditions. For these reasons, we concluded that ssDUE recruitment mechanism conducted by HBsu may be a major pathway of initiation in B. subtilis. In the revised manuscript, we added new sentences related to this, as below.
Line 148-152 in the revised manuscript; In B. subtilis, HU homologue HBsu is essential for replication initiation from oriC, which is consistent with the idea that DNA bending is essential for initiation from oriC and thus ssDUE recruitment mechanism could be conserved in B. subtilis [59]. Unwinding of B. subtilis oriC in supercoiled DNA should be analyzed in the presence of HBsu.
3) Lines 261-275, Fig 2C: According to published literature, R1 is bound by DnaA throughout the cell cycle (reported in ref 16, 19, and 62). Given this, if binding of DnaA to R1 prevents binding of IHF to IBS2, it is hard to see how IBS2 affect binding at IBS1 at any time, or be relevant to initiation timing. The authors should at the least revise this section to include whether persistent R1 binding would reduce the role of IBS2, or remove the section and Fig 2C.
--Thank you for the comment about the DnaA dynamics at R1 and we basically agree with you that DnaA protein can bind to R1 throughout cell cycle, but our current study suggested that specifically at the initiation period, IHF covers IBS2 which overlaps with R1. Because the reference 16, 19, and 69 in the revised manuscript (16, 19, and 62 in the original manuscript) only qualitatively showed DnaA-R1 binding but none of them quantitatively analyzed the proportion of DnaA-R1 binding, the possibility of temporal IHF-IBS2 binding cannot be excluded. Thus, our constructive discussion about the significance of IBS2 can NOT be removed or changed.
4) Line 345: The appropriate (or at least earlier) reference for mioC transcription in the cell cycle is Theisen et al., Mol Microbiol, 10: 575, 1993.
--Thank you for pointing this out, and we added the reference.
5) Section 3.2.3, Line 239: there is a brief statement that IHF and Fis could promote specific DnaA interaction within DARS2, but it is unclear whether there are any published studies that might clarify the roles of the binding proteins at DARS2. Have the sites been mutated? A slightly more detailed discussion, even if speculative, could increase the impact of the manuscript.:
-- The reviewer raises a very good point here and the detailed mechanism of DARS2 activation is still mystery. To increase the impact of this review, we included the sentence about the previous studies about DARS2 IBS/FBS mutants which inactivate DARS2 function (Kasho et al., NAR, 2014, 42, 13134-13149).
Very minor comment: Line 120, the word “strand” is misspelled.
--Thank you for pointing this out, and we corrected the word.

Author Response
Response to Reviewer 2's comments
This is a well written and comprehensive review highlighting the importance of binding/dissociation of DNA architectural proteins in the control of DNA replication initiation. In this area, majority of our knowledge has come from the heroic work of Katayama lab. Hence, this is an authoritative review. Despite the progress, the review has emphasized unabashedly how much remains to be done. The review is thus also futuristic. I have made a few minor comments towards improving the clarity of presentation. Since this is a review, it is quite appropriate to speculate a bit on possible mechanisms when the results appear confusing (comment 7 below).
--We appreciate the reviewer for the assessment of our manuscript. All revisions are highlighted in red characters in the revised manuscript.Please see the attachment.
I am also curious whether in the grand scheme of things acidic phospholipid-mediated rejuvenation of DnaA play a significant role.
--We are of course interested in the role of acidic phospholipids, and we briefly include a sentence about acidic phospholipid system and citation in Chapter 3-2-3.
Line 259-262 in the revised manuscript: As an additional note, acidic phospholipids are suggested to mediate ATP-DnaA production from ADP-DnaA; however, its biological significance as well as the cell cycle-coordinated regulation remain to be further explored [94].
- 10: Delete representative ie --two nucleoid associated proteins---. The nucleoid has thousands of proteins. Calling the two NAPS as representative may not be justified. Delete representative also from l.43.
--Thank you for pointing this out, the words 'representative' in Line 10 and 43 are removed in the revised manuscript. - 35: I will say chromosome and NOT genome throughout. If cells contained plasmids or secondary chromosomes, then use of genome would make sense. The use of genome is acceptable in l.68.
--Thank you for your comment and the words 'genome' are replaced to 'chromosome' in Line. 35, 38, 44, 46, and 415 in the revised manuscript. - 48: Delete 13-mer. The term should be reserved for the three 13-mers of oriC to avoid confusion.
--Thank you for your comment. To avoid the confusion, the word '13-mer' in Line 48 is removed in the revised manuscript. - 78: AT-cluster region needs to be introduced here as well as in the Figure legend. It is there much later (l.142).
--Thank you for your comment and now we moved the sentence about 'AT-cluster regions that flanks the DUE just outside of the minimal oriC region' from Line 142 to Line 78, and instead the explanation in Line 160 in the revised manuscript (Line 142 in the original manuscript) was simplified. Also additional explanation about AT-cluster is included in the legend of Figure 1C (Line 453-454). - 87-97: The authors may cite that the putative roles of domain II are beyond placing domains I and III correctly (Hou, Y.---Saxena, R. 2022 Sci. Adv. 8:eabq6657), and the role of His136 in DnaA-DnaA oligomerization (Saxena et al 2020 NAR 48:200).
--Thank you for your suggestion. We of course took Saxena et al. (2020, NAR and 2022, Sci Adv) into considerations, but the evidence is limited to specific mutants of DnaA; i.e. 1) in Saxena et al. (2020, NAR, 48:200), DnaA domain II mutant D118Q, but not D118A cause defect in ATP binding and proper replication initiation at oriC, and 2) in Hou et al.(2020, Sci Adv, 8:eabq665) DnaA H136Q, but not H136A causes DnaA oligomerization defect. Notably, in Sakiyama et al. (2018, Front Microbiol, 9:2017), DnaA H136A mutant sustains oriC-DnaA complex formation and DUE unwinding activity. Thus, based on these limited analyses, we included additional explanation related to this as well as its reference in revised manuscript.
Line 93-96 in the revised manuscript; Recent studies have highlighted the specific role of the His136 residue of domain III in DnaB loading onto ssDUE, consistent with the structural insight that DnaA His 136 residue is exposed to the surface and potentially interacts with other proteins [26,27,42]. - 99: Delete reference to Fig. 1C.
--The reference to Figure 1C is deleted in the revised manuscript. Thank you. - 100: Why no t1 in Fig. 1C or 2A?
--Thank you for pointing out the lack of tau1. Figures 1A and 2A in the revised manuscript include tau1.
- 115: How does R2 help in DnaA assembly. Even a speculative statement would help. Since R2 is oriented, it is not obvious how the two oppositely oriented DnaA-box clusters are both helped. Does R2 work if inverted?
--Thank you for the important comment. In the revised manuscript we added the mechanistic view of R2-mediated DnaA assembly at R-DOR as well as its reference 45: Shimizu et al., (PNAS, 2016, 113, E8021-E8030).
The addition is : --- potentially via domain I-mediated DnaA-DnaA interaction [45,48]. Spaces from R2 to I2 and C3 are long enough to prevent domain III-domain III interaction. (Line 119 in the revised manuscript)
About R2 inversion, Langer et al. (Mol Microbiol, 1996, 21, 301-311) analyzed oriC R2 inversion/scrambled mutant on plasmid and suggested that those R2 mutations moderately inhibit minichromosome replication. Also, Weigel et al. (Mol Microbiol, 2001, 40, 498-507) suggested that chromosomal oriC R2 scrambled mutation very slightly inhibits the initiation. Those are consistent with a stimulatory role for R2. - 151: Regulation of? Also in L.152.
--The typos are corrected in the revised manuscript, thanks. - 152, 168, 193 & 219: Add ‘The’ ie The Sequestration System etc.
--Thank you, we corrected in the revised manuscript. - 161: There are only 9 red stars shown in Fig. 2A.
--Thank you very much for pointing out the mistake. We corrected Figure 2A and that has eleven GATC sites in the revised manuscript. - 178: constructed or assembled?
--Thank you, we corrected in the revised manuscript. - 204: Is this a contradiction-IHF-mediated DnaA-DnaA interaction between R1 and R5 also happens at oriC.
--Thank you for pointing out the confusing sentence, now we corrected the sentence as below in the revised manuscript, and resolve the contradiction that IHF-mediated DnaA-DnaA interaction between R1 and R5 does NOT induce DDAH.
Line 222 in the revised manuscript: Thus, IHF-dependent DnaA-DnaA interactions at datA locus specifically support DDAH. - 231: Delete “Unlike---datA” -datA also has head-to-head DnaA boxes.
--Thank you, we deleted 'and datA' in the revised manuscript. - 252: 26-TT-27 or 25-TT-26?
--Thank you very much for pointing out the mistake. We corrected 25-TT-26 in the revised manuscript (Line 273). - 277-280: Delete. Unnecessary and confusing when you are discussing Fis-bindingwithin a cell-cycle.
--Thank you for your comment. As you suggested the paragraph is confusing, but explanation about Fis expression and DARS2-Fis binding at exponential growth phase is consistent with the timely DARS2 activation in rapidly-growing cells. Thus we concluded that these sentences are necessary. To avoid the confusion, we moved the paragraph to the end of this section with minor revision (Line 311-314 in the revised manuscript).
- 289-294: Rearrange. The statement on DiaA is interrupting the theme of negative feedback.The statement starting with “This simple negative feedback—” should follow the negative feedback on l.289.
--Thank you for pointing out the confusing arrangement of the paragraph, according to your suggestion we changed the order of these sentences (Line 308-310 in the revised manuscript) - 301: Replace Miyoshi et al with [85].
--The reference style is corrected in the revised manuscript, thanks. - 3 needs a new legend. Now it has the same legend as in Fig.2. Author contribution says “all authors have read”.
--Thank you very much for pointing out the mistake. The revised manuscript has a new legend.

Round 2
Reviewer 1 Report
The revised version of this manuscript remains incomplete, because the authors chose to not adequately address the issues raised in the review of the original manuscript. Their reasons for not revising the manuscript are not convincing. As currently written, the manuscript offers only an unbalanced and incomplete review of the topic, and so remains unsuitable for publication.
1. The authors state “In addition, previous studies suggested that Fis can bind to a specific site between R2 and C3 in the chromosomal oriC but this specific oriC-Fis binding does not influence the initiation 122 of replication [20,50,51], although there remains a possibility that the binding could slightly influence initiation depending on growth conditions [52,53]. Further experiments are essential to prove the regulatory role of Fis at oriC.” A valid review article can’t ignore published studies because there remain some questions. (If this were the case, then much of the material regarding IHF and Fis activity at the DARS2 locus, as well as the role of IBS2, would have to be excluded). In the case of oriC, the authors fail to consider published literature where a mutation in the primary Fis binding site does have a slight effect on initiation. Further, the authors do not consider any of the published evidence from multiple labs that Fis can bind to more than one site in oriC, and binding to these sites may exert an effect on formation of the initiation complex. One should also note that mutations knocking out IHF binding to its site in oriC have even less effect on initiation than knocking out the “primary” Fis site. For IHF “further experiments” did suggest that this is probably due to the fact that the IHF binding region is intrinsically bendable and will bind HU or some other NAP in the absence of IHF. Similarly, in vivo there could be factors mitigating loss of Fis binding to one of its oriC sites.
2. The authors did not comply with the Reviewer’s request to include the dynamic Fis/oriC interaction during the cell cycle and models describing how this affects the timely assembly of the initiation complex in a growth-rate dependent way. This makes the review unacceptably unbalanced and incomplete.
3. In the rebuttal, the authors state that the oriC section is not the main part of the review as a reason for not including the role of Fis in regulating IHF binding and thus in formation of the initiation complex. However, the oriC material has equal billing in the title and the abstract, and the words regarding oriC occupy slightly more than half the manuscript.
4. The authors did not adequately address the Reviewer’s request to clarify what exactly they mean by the ss recruitment model, and thus the manuscript is still confusing with regard to other bacterial origins. The authors now include a reference to recent data showing that HU/DNA bending is needed for initiation in B. subtillis, but neglect to state that there is no evidence that HU is bending origin DNA to put the DUE near a bound DNA-ATP oligomer. Some further explanation of where bending must be positioned in the origin to be consistent with the ss recruitment model is needed.
5. The authors did not comply with the request to include a brief discussion whether persistent R1 binding would reduce the role of IBS2 in regulating binding of IBS1. If the authors really think that it is probable that DnaA transiently dissociates from R1 in a small subset of cells at some point during the cell cycle, and that this could somehow affects IHF binding to either of its putative sites, then they should have provided a lucid explanation of how this could be.
Author Response
The revised version of this manuscript remains incomplete, because the authors chose to not adequately address the issues raised in the review of the original manuscript. Their reasons for not revising the manuscript are not convincing. As currently written, the manuscript offers only an unbalanced and incomplete review of the topic, and so remains unsuitable for publication.
> We adequately revised the original manuscript considering all of the previous suggestions. Unfortunately, this comment refers to our response only about the oriC-Fis interaction. However, even for this issue, we have carefully considered all previous reports and have included them in our comments and the revised manuscript, ensuring comprehensive coverage of this interaction. Notably, Reviewer 1 acknowledged in their comment that a mutation in the primary Fis binding site of oriC has a slight effect on initiation, supporting our viewpoint on the minor role of oriC-Fis in regulation, which was concordantly indicated by three independent laboratories based on careful in vivo and in vitro analyses. Thus, the manuscript is overall described with fair, balanced, and comprehensive views. Therefore, this reviewer’s comment is not valid. Also, see below for more details.
- The authors state “In addition, previous studies suggested that Fis can bind to a specific site between R2 and C3 in the chromosomal oriC but this specific oriC-Fis binding does not influence the initiation 122 of replication [20,50,51], although there remains a possibility that the binding could slightly influence initiation depending on growth conditions [52,53]. Further experiments are essential to prove the regulatory role of Fis at oriC.” A valid review article can’t ignore published studies because there remain some questions. (If this were the case, then much of the material regarding IHF and Fis activity at the DARS2 locus, as well as the role of IBS2, would have to be excluded).
> As mentioned above, we have carefully considered all previous reports about the oriC-Fis interaction and have included them in our comments and the revised manuscript.
About DARS2 activation, we showed direct evidence of the interactions in vivo and clearly revealed that IHF and Fis are essential for DARS2 function in vivo and in vitro (Kasho et al., Nucleic Acids Res, 2014, 42:13134-13149.). These results well coincide with previous studies of the fis mutant phenotypes as described in the present review manuscript. No study opposing these finding has been reported. Also about IHF binding at oriC IBS2, we include a result that at the initiation period, IHF binds to not only IBS1 but covers IBS2 region which overlaps with R1 (Kasho et al., Front Microbiol, 2021, 12, 697712.). Thus, there is no reasons to exclude the sentences about DARS2 IBS/FBS and oriC IBS2.
In the case of oriC, the authors fail to consider published literature where a mutation in the primary Fis binding site does have a slight effect on initiation. Further, the authors do not consider any of the published evidence from multiple labs that Fis can bind to more than one site in oriC, and binding to these sites may exert an effect on formation of the initiation complex.
> As we wrote in the first round of our comments, we have carefully considered all previous reports about the oriC-Fis interaction, and have included them in our comments; i.e. Weigel et al (Mol Microbiol, 2001, 40, 498-507), Margulies and Kaguni (J Biol Chem, 1998, 26, 5170-5175), (Nucleic Acids Res, 1996, 24, 3527-3532), Ryan et al. (Mol Microbiol, 2004, 51, 1347-1359), Cassler et al., (EMBO J, 1995, 14, 5833-5841), Kaur et al. (Mol Microbiol, 2014, 91, 1148-1163), Rao et al. (Front Microbiol, 2018, 9:1673). Taken these conflicting reports all together, we fairly judged that the conflicts are not resolved so far, and further analysis and direct evidence are still required for proving the significance of chromosomal oriC-Fis binding. Notably, Reviewer 1 acknowledged in their comment that a mutation in the primary Fis binding site of oriC has a slight effect on initiation, supporting our viewpoint on the minor role of oriC-Fis in regulation. In addition, in vivo evidence of the multiple Fis binding to oriC has not been shown and such binding was observed only in vitro by extremely excessive supply of Fis. We thus conclude that these are too preliminary to be included in this short review. As such, the manuscript was written with fair, balanced and comprehensive views. Also, see our response below.
One should also note that mutations knocking out IHF binding to its site in oriC have even less effect on initiation than knocking out the “primary” Fis site. For IHF “further experiments” did suggest that this is probably due to the fact that the IHF binding region is intrinsically bendable and will bind HU or some other NAP in the absence of IHF. Similarly, in vivo there could be factors mitigating loss of Fis binding to one of its oriC sites.
> DNA bending at oriC by IHF/HU is essential for replication initiation. Our current study (Yoshida et al., Nucleic Acids Res 2023, 51, 6286–6306) clearly revealed that in the absence of IHF, HU binds to oriC IBS to stimulate initiation. Consistently, as Reviewer 1 wrote, HU can potentially bind to the mutated oriC IBS. About Fis, Cassler et al. (EMBO J, 1995, 14, 5833-5841) performed in vivo DMS footprinting with minichromosome and provided the data that Fis only faintly binds to oriC throughout whole cell cycle, supporting our conclusion that Fis plays only a minor function at oriC and initiation regulation is mainly achieved by ATP-DnaA oligomerization. In addition, as described in the revised manuscript, in vivo analysis of this issue still remains to be further investigated. Thus, we revised the related sentence with more careful expression. In addition, we refer to a most recent comprehensive review describing the suggested role of Fis on oriC (Grimwade and Leonard, Front Micro., 2021), as described below and in the revised manuscript (highlighted with yellow background). It is important to note that no new reports on the Fis-oriC interaction have been published since the release of that review.
Line 120-124; In addition, previous studies suggested that Fis can bind to a specific site between R2 and C3 in the chromosomal oriC but this specific oriC-Fis binding does not or may only slightly influence the initiation of replication [20,50,51,52,53]. A recent review summarizes roles and mechanisms suggested for Fis-oriC interaction [7].
- The authors did not comply with the Reviewer’s request to include the dynamic Fis/oriC interaction during the cell cycle and models describing how this affects the timely assembly of the initiation complex in a growth-rate dependent way. This makes the review unacceptably unbalanced and incomplete.
> We agree with the referee in that Fis expression is regulated in a manner dependent on growth rate. However, dynamic Fis/oriC interaction during the cell cycle has not been demonstrated and thus this issue still remains to be further investigated and is too preliminary to be included in this short review. Instead, we refer to a most recent comprehensive review describing the suggested role of Fis on oriC (Grimwade and Leonard, Front Micro., 2021,12:732270.).
- In the rebuttal, the authors state that the oriC section is not the main part of the review as a reason for not including the role of Fis in regulating IHF binding and thus in formation of the initiation complex. However, the oriC material has equal billing in the title and the abstract, and the words regarding oriC occupy slightly more than half the manuscript.
> The mechanism of replication initiation at oriC has been well studied and the updated information is necessary to understand the central topics of this manuscript.
- The authors did not adequately address the Reviewer’s request to clarify what exactly they mean by the ss recruitment model, and thus the manuscript is still confusing with regard to other bacterial origins. The authors now include a reference to recent data showing that HU/DNA bending is needed for initiation in B. subtillis, but neglect to state that there is no evidence that HU is bending origin DNA to put the DUE near a bound DNA-ATP oligomer. Some further explanation of where bending must be positioned in the origin to be consistent with the ss recruitment model is needed.
> As we wrote in the first round of our comments, Karaboja and Wang (J Bacteriol, 2022, 204, e0011922) clearly indicated that in B. subtilis, IHF/HU homologue HBsu is essential for replication initiation from oriC, well consistent with the idea that DNA bending is essential for initiation from oriC and ssDUE recruitment mechanism is present in B. subtilis. Also, Krause et al. (J. Mol. Biol. 1997, 274, 365-380) provided results that E. coli HU stimulates DUE unwinding of B. subtilis oriC in vitro, supporting our idea that ssDUE recruitment model may be a major pathway of initiation in B. subtilis, although naturally, in B. subtilis, interaction of HBsu with oriC has not been demonstrated. We revised the related sentence including these, as described below and in the revised manuscript (highlighted with yellow background).
Line 147-149; In B. subtilis, HU homologue HBsu is essential for in vivo replication initiation from oriC and E. coli HU stimulates in vitro DUE unwinding of B. subtilis oriC [59, 60].
Line 632-633; [60] Krause, M.; Ru, B.; Lurz, R.; Messer, W. Complexes at the Replication Origin of Bacillus subtilis with Homologous and Heterologous DnaA Protein. J. Molculer Biol. 1997, 274, 365–380.
- The authors did not comply with the request to include a brief discussion whether persistent R1 binding would reduce the role of IBS2 in regulating binding of IBS1. If the authors really think that it is probable that DnaA transiently dissociates from R1 in a small subset of cells at some point during the cell cycle, and that this could somehow affects IHF binding to either of its putative sites, then they should have provided a lucid explanation of how this could be.
> As we wrote in the first round of our comments, our current study suggested that specifically at the initiation period, IHF covers IBS2 which overlaps with R1. Because the reference 16, 19, and 69 in the revised manuscript only qualitatively showed DnaA-R1 binding but none of them quantitatively analyzed the proportion of DnaA-R1 binding, the possibility of temporal IHF-IBS2 binding cannot be excluded.
